# Resurrection of the ancestral RH5 invasion ligand provides a molecular explanation for the origin of *P. falciparum* malaria in humans

**Francis Galaway**[1], **Ryan Yu**[1], **Anastasia Constantinou**[1], **Franck Prugnolle**[2]*, **Gavin J. Wright**[1]*

**1** Cell Surface Signalling Laboratory, Wellcome Sanger Institute, Cambridge, United Kingdom, **2** Laboratoire MIVEGEC, Univ Montpellier, UMR CNRS 5290-IRD224-UM, Montpellier, France

* gw2@sanger.ac.uk (GJW); franck.prugnolle@ird.fr (FP)

**Data Availability Statement:** All relevant data are within the paper and its Supporting Information files.

## Abstract

Many important infectious diseases are the result of zoonoses, in which pathogens that normally infect animals acquire mutations that enable the breaching of species barriers to permit the infection of humans. Our understanding of the molecular events that enable host switching are often limited, and yet this is a fundamentally important question. *Plasmodium falciparum*, the etiological agent of severe human malaria, evolved following a zoonotic transfer of parasites from gorillas. One gene—*rh5*—which encodes an essential ligand for the invasion of host erythrocytes, is suspected to have played a critical role in this host switch. Genome comparisons revealed an introgressed sequence in the ancestor of *P. falciparum* containing *rh5*, which likely allowed the ancestral parasites to infect both gorilla and human erythrocytes. To test this hypothesis, we resurrected the ancestral introgressed reticulocyte-binding protein homologue 5 (RH5) sequence and used quantitative protein interaction assays to demonstrate that this ancestral protein could bind the basigin receptor from both humans and gorillas. We also showed that this promiscuous receptor binding phenotype of RH5 was shared with the parasite clade that transferred its genome segment to the ancestor of *P. falciparum*, while the other lineages exhibit host-specific receptor binding, confirming the central importance of this introgression event for *Plasmodium* host switching. Finally, since its transfer to humans, *P. falciparum*, and also the RH5 ligand, have evolved a strong human specificity. We show that this subsequent restriction to humans can be attributed to a single amino acid mutation in the RH5 sequence. Our findings reveal a molecular pathway for the origin and evolution of human *P. falciparum* malaria and may inform molecular surveillance to predict future zoonoses.

## Introduction

The majority of emerging infectious diseases are zoonotic and arise by the acquisition of mutations that permit the infection of humans [1]. Notable examples are viruses such as influenza and HIV, but also include parasites such as *P. falciparum*, which causes the most severe form

**Funding:** This work was funded by the Wellcome Trust (https://wellcome.ac.uk/) grant 206194 (GJW) and Labex Centre Méditerranéen de l'Environnement et de la Biodiversité (http://www.labex-cemeb.org/) (FP). The funders had no role in study design, data collection and analysis, decision to publish, or preparation of the manuscript.

**Competing interests:** The authors have declared that no competing interests exist.

**Abbreviations:** AVEXIS, avidity-based extracellular interaction screening; CyRPA, cysteine-rich protective antigen; GTR, General Time Reversible; HEL, human erythroid-like; IntRH5, introgressed ancestral RH5; MSP, merozoite surface protein; PE, phycoerythrin; Rh, reticulocyte-binding homologue; RH5, reticulocyte-binding protein homologue 5; RIPR, RH5-interacting protein; SPR, surface plasmon resonance.

of malaria [2]. *Plasmodium* parasites have evolved to infect many different species of animal, including mammals, birds, and reptiles, and are often restricted to their known hosts [3]. While ecological factors are likely to play an important role in *Plasmodium* host restriction in the wild, laboratory experiments show that molecular barriers also exist [4]. Our understanding of the molecular pathways involved in *Plasmodium* host switching are not well understood but could be used to help predict and preempt future zoonoses that may result in deadly pandemics.

*P. falciparum* malaria is a devastating infectious disease that is one of the world's major health problems and is still responsible for over 400,000 deaths annually [5]. The origin of *P. falciparum* was recently shown to be the *Laverania*: a subgenus of parasites that infect African great apes, and which exhibit a strict tropism for their hosts. Beyond *P. falciparum*, the *Laverania* contains six other species, of which three, *P. gaboni*, *P. reichenowi*, and *P. billcollinsi*, infect only chimpanzees; and the remaining three, *P. adleri*, *P. blacklocki*, and *P. praefalciparum*, exclusively infect gorillas [6,7]. *P. falciparum* is most closely related to the gorilla-restricted parasite *P. praefalciparum*, diverging from their most recent common ancestor between forty and sixty thousand years ago, at around the same time as the expansion of humans out of Africa [8]. Phylogenetically, the *Laverania* separate into two distinct clades (A and B), and genome comparisons highlighted a rare inter-clade introgression of an approximately 8-kbp genomic region from an ancestor of *P. adleri* to the precursor of *P. falciparum*/*P. praefalciparum* [8,9] (Fig 1A). Introgressions are the transfer of genomic regions from one species into another, caused by hybridization followed by recurrent crossing between the hybrid and its descendants and the parental species. The introgressed region encoded two members of a protein complex that is essential for invasion of erythrocytes: reticulocyte-binding protein homologue 5 (RH5) and cysteine-rich protective antigen (CyRPA) (S1 Fig), suggesting it may have been important in the zoonotic origin of *P. falciparum*.

RH5 is a member of the *P. falciparum* reticulocyte-binding homologue (Rh) family of ligands, which are released by the parasite during invasion and are required for host cell recognition by directly interacting with receptor proteins displayed on the erythrocyte surface [10]. RH5 differs from the other members of the Rh family because it encodes a secreted rather than membrane-tethered protein and cannot be genetically deleted in any *P. falciparum* strain, demonstrating that it is essential for parasite growth in blood stage culture [10]. Once released at the point of invasion, RH5 is tethered to the parasite surface by direct interactions with the membrane-anchored protein P113 [11]. RH5 was found to be part of a larger protein complex, and immunoprecipitation followed by mass spectrometry identified two other secreted parasite proteins: CyRPA [12] and RH5-interacting protein (RIPR) [13] (Fig 1B). Subsequent biochemical binding studies [11] and then structural work [14] on the complex have revealed that only CyRPA bound both RH5 and RIPR, demonstrating that it acts as a central organizer of the complex. Using an assay developed to detect extracellular protein interactions [15], the host erythrocyte receptor for RH5 was identified as basigin, and this interaction is both essential and universally required by all strains of *P. falciparum* for invasion [16], making RH5 and the other components of the complex important blood stage vaccine targets [17].

One puzzling evolutionary aspect of the RH5 invasion complex is that obvious sequence orthologues of each component do not co-occur in the genomes of sequenced *Plasmodium* genomes. Genes encoding P113 and RIPR are present in both primate and rodent-infecting parasite species, while *CyRPA* is present in primate but not rodent parasites. RH5 and the use of basigin as an essential erythrocyte invasion receptor appear to be recent evolutionary innovations because genes encoding RH5 proteins are restricted to the *Laverania* and basigin is not essentially required for erythrocyte invasion by either *P. vivax* or *P. knowlesi* [18]. Variants of RH5 in *P. falciparum* have been shown to have roles in host-specific invasion [19,20], and the

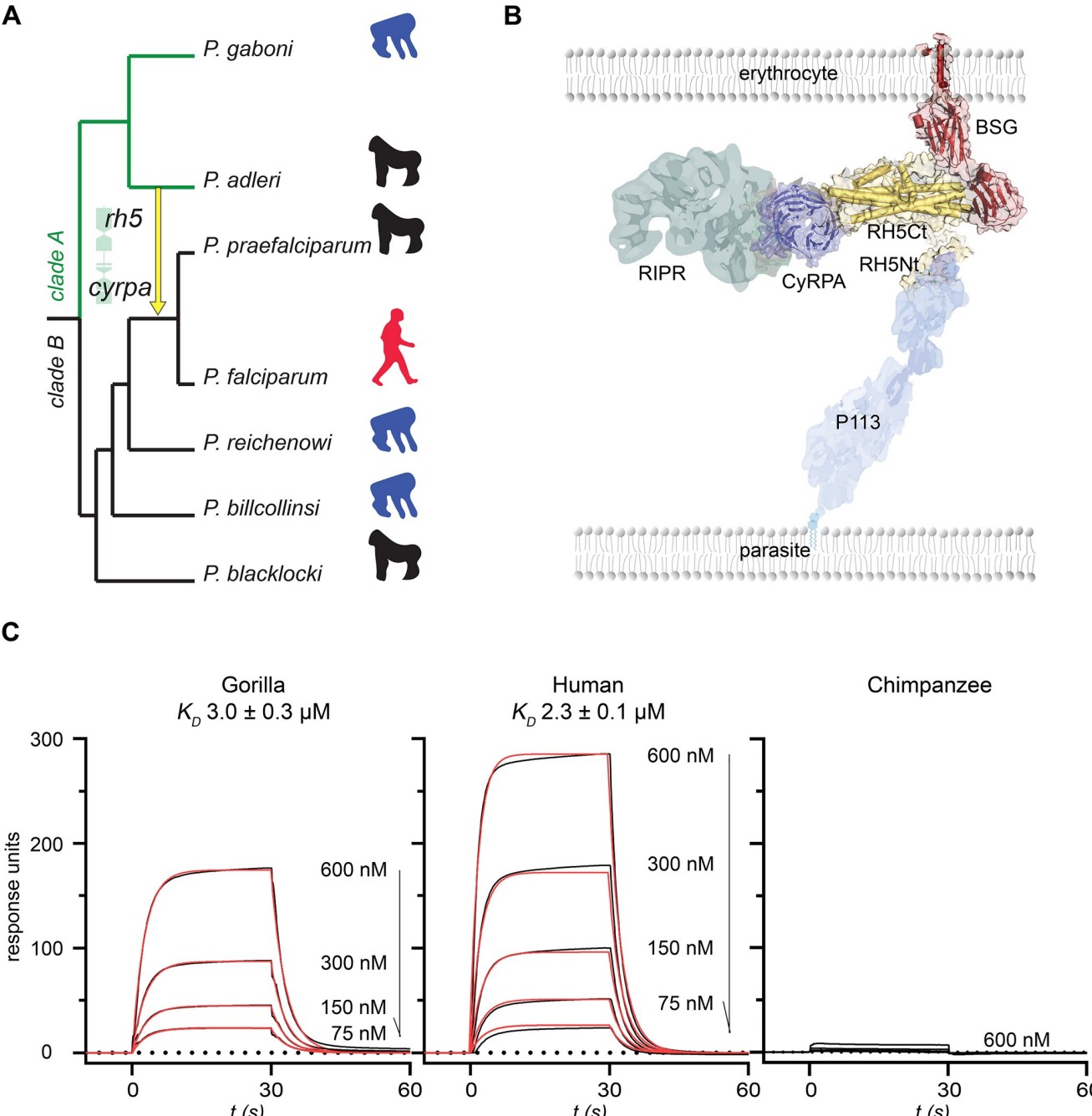

**Fig 1. RH5 from an ancient inter-clade introgression in *Laverania* parasites binds both human and gorilla basigin. (A)** Phylogenetic relationships and host tropisms of the parasites within the *Laverania* subgenus. The inter-clade introgression event encoding *rh5* and *cyrpa* is marked with a yellow arrow. Hosts are indicated as chimpanzee (blue), gorilla (black), and human (red) icons. **(B)** Organization of the RH5 invasion complex. RH5 is tethered through its N terminus (RH5Nt) to the parasite surface via P113 and binds host basigin on the erythrocyte surface; CyRPA directly interacts with RH5, thereby recruiting RIPR to the complex. **(C)** Ancestral introgressed RH5 binds to gorilla and human, but not chimpanzee, basigin. Serial dilutions of the indicated concentrations of purified introgressed RH5 were injected over gorilla, human, and chimpanzee basigin immobilized on a sensor chip. Raw surface plasmon resonance (SPR) traces are shown in black and fitted to a simple 1:1 Langmuir binding isotherm (red) to derive the equilibrium dissociation constants ($K_D$s). Underlying numerical data can be found in S1 Data. A representative experiment of three is shown. BSG, basigin; *cyrpa*, cysteine-rich protective antigen; RH5, reticulocyte-binding protein homologue 5; RH5Ct, RH5 C-terminal domain; RH5Nt, RH5 N-terminal domain; RIPR, RH5-interacting protein; SPR, surface plasmon resonance.

inability of *P. falciparum* RH5 to bind gorilla and only weakly to chimpanzee basigin suggested the RH5–basigin interaction was responsible for the restriction of *P. falciparum* to humans [21]. It was widely thought, therefore, that host restriction in Laveranian parasites was determined by the specificity of the RH5–basigin interaction and that the transfer of this gene from *P. adleri* into the ancestor of *P. falciparum*/*P. praefalciparum* was the founding event leading to the evolution of one of the most deadly human parasites, *P. falciparum* [8].

By comparing the binding properties of extant and calculated ancestral sequences of the RH5 invasion complex within the *Laverania* subgenus, here we show that a crucial property of the introgressed RH5 was the ability to bind both gorilla and human basigin receptors. Using mutational analysis, we demonstrate that a single residue in RH5 can explain the subsequent restriction of *P. falciparum* to humans, thereby revealing a molecular pathway for the zoonotic origin of *P. falciparum* malaria.

## Results

To investigate how the inter-clade transfer of genetic material encoding RH5 contributed to the zoonotic origin of *P. falciparum* in humans, we determined, using probabilistic-based approaches and the known sequences of the extant species, the likely sequence of the introgressed ancestral *rh5* gene. Our analysis converged on a single protein sequence (introgressed ancestral RH5 [IntRH5]) with very high confidence at each residue (S1 Table). A synthetic gene corresponding to this sequence was constructed, and expressed as a recombinant protein in mammalian cells (S2 Fig). The protein was purified and the biophysical binding parameters to human, gorilla, and chimpanzee basigin were quantified using surface plasmon resonance (SPR). Remarkably, we observed that the introgressed RH5 bound both human and gorilla basigin with comparable affinities (Fig 1C); no binding was detected to chimpanzee basigin. We confirmed the ability of the introgressed RH5 protein to specifically bind human basigin in the context of a cell membrane by using a well-characterized RH5 cell surface binding assay [22] (S3 Fig). Biotinylated RH5 was clustered around a fluorescent streptavidin conjugate to create a highly avid fluorescent binding probe, which was incubated with human erythroid-like (HEL) cells. The RH5 probe bound cell surface basigin, causing a shift in the fluorescence signal when analyzed by flow cytometry; the binding specificity for the basigin receptor was determined by preincubating the cells with an anti-basigin monoclonal antibody that prevented binding. This promiscuous binding tropism is in striking contrast to the restriction of *P. falciparum* RH5 for human basigin [21] and suggested a plausible mechanism for the zoonotic infection of humans by a gorilla parasite.

We next asked if this promiscuous host receptor binding was a unique property of the ancestral introgressed RH5 by determining the RH5–basigin receptor–ligand tropisms across the extant *Laverania* parasites. We cloned, expressed, and purified the RH5 orthologues from six *Laverania* parasites (S2 Fig) and systematically quantified their binding affinities to human, gorilla, and chimpanzee basigin. The RH5 orthologues from all *Laverania* parasites bound their respective host basigin receptors, showing that the RH5–basigin interaction is conserved across the *Laverania* and that the proteins were correctly folded and functional (Fig 2A). Like *P. falciparum*, the RH5 proteins from the clade B parasites *P. reichenowi* and *P. billcollinsi* bound only to the basigin receptor from their known host (Fig 2A). However, the RH5 proteins from *P. praefalciparum* and both clade A parasites *P. gaboni* and *P. adleri*, while binding the basigin receptor from their known ape host, also exhibited the same promiscuous binding profile as the introgressed RH5 by additionally binding human basigin (Fig 2A). Again, we confirmed these findings using the cell-based RH5 binding assay (Fig 2B).

During erythrocyte invasion, RH5 interacts with components of a larger protein complex that includes the membrane-tethered P113 protein and two other secreted proteins: CyRPA

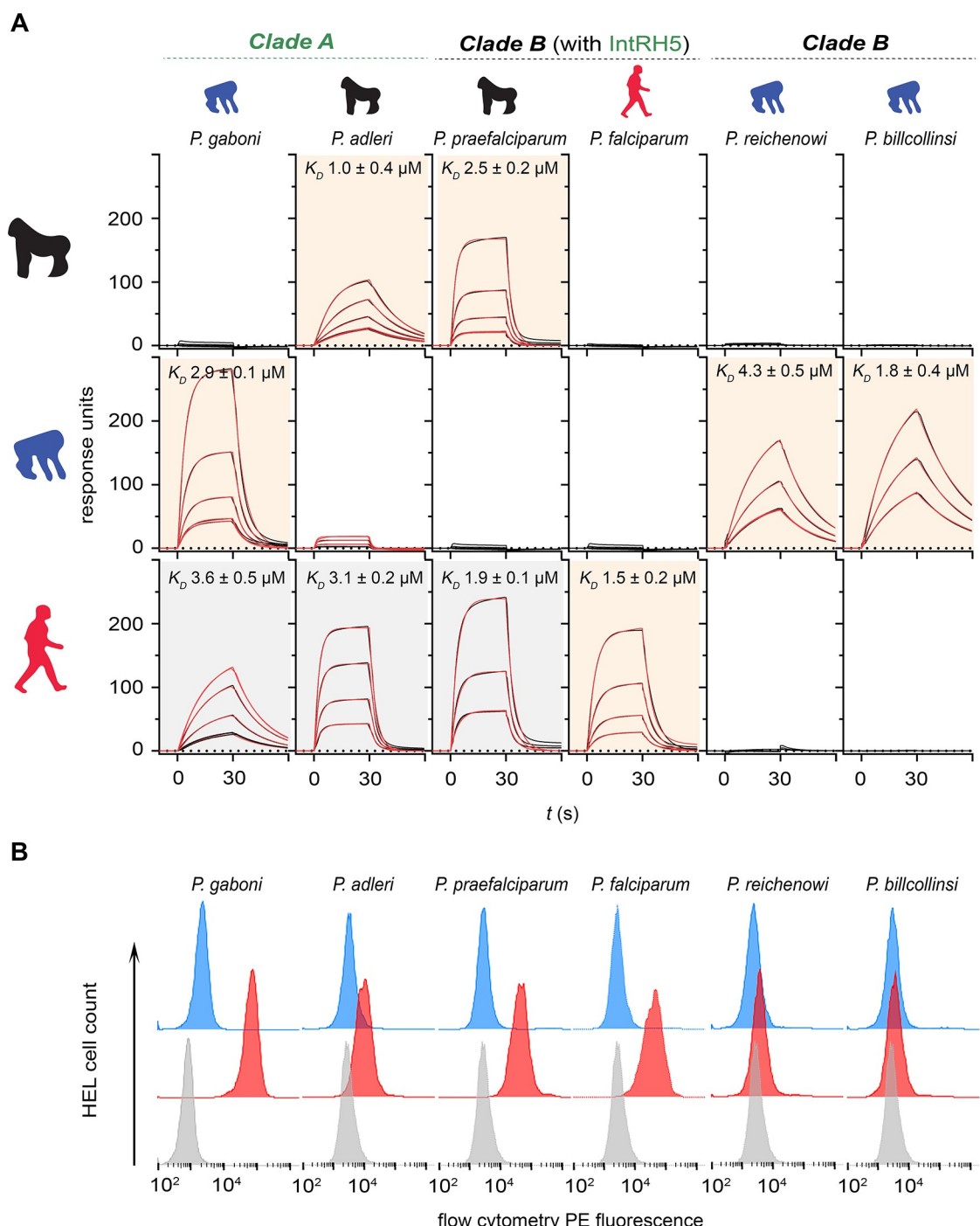

**Fig 2. *P. praefalciparum* and clade A *Laverania* RH5 does not exhibit strict host basigin binding specificity. (A)** The binding of *Laverania* RH5 proteins to human, gorilla, and chimpanzee basigin was quantified by SPR. Sensorgram data from serial dilutions of the RH5 proteins are shown in black and were fitted to a simple 1:1 Langmuir binding isotherm (red). Background shading of the graphs indicates where binding was expected based on the known parasite tropism (beige) or were unexpected (gray). **(B)** Binding specificities of *Laverania* parasite RH5 proteins with human basigin were confirmed using an RH5 cell binding assay. Avid RH5 binding probes were presented to basigin-expressing HEL cells, and binding specificity was demonstrated by showing that all RH5 probe binding was abolished if the cells were preincubated with an anti-basigin monoclonal antibody (mAb) that blocks RH5 binding (Ab1—blue histograms) compared with a cell-binding isotype-matched anti-CD59 mAb (red). Control is streptavidin-PE alone (gray). Summary numerical data are provided in S1 Data; gating strategy and original .fcs files in S2 Data. Representative results of at least three independent experiments are shown. Ab1, anti-basigin mAb; HEL, human erythroid-like; IntRH5, introgressed ancestral RH5; mAb, monoclonal antibody; PE, phycoerythrin; RH5, reticulocyte-binding protein homologue 5; SPR, surface plasmon resonance.

and RIPR (Fig 1B). Interestingly, the introgressed fragment encompassed two genes encoding RH5 complex components: RH5 and CyRPA, which are genetically closely linked (S1 Fig). Was the cotransfer of the gene encoding CyRPA an important factor that permitted human infection? And immediately following the introgression event, could the different components interact to form a functional complex? To address these questions, we again used ancestral sequence reconstruction to determine the sequences of the introgressed CyRPA and ancestral P113 and RIPR (S1 Table). We expressed these proteins as enzymatically monobiotinylated "baits" and multimerized enzyme-tagged "preys" and tested them for the ability to directly interact using an assay called avidity-based extracellular interaction screening (AVEXIS), which is specifically designed to detect extracellular protein interactions [15]. Using this approach, we showed that the introgressed RH5 could interact with both the introgressed CyRPA and ancestral P113 and that the introgressed CyRPA could bind ancestral RIPR (Fig 3A). We further showed that the proteins interacted directly and determined their biophysical parameters using SPR (S4 Fig). These results are consistent with the known binding properties of the RH5 complex components [11] and their structural arrangement [14], demonstrating that RH5 and CyRPA encoded on the introgressed fragment could have formed a functional RH5 complex.

The gene encoding RH5—although not the other members of the complex—is restricted to the *Laverania* subgenus, suggesting that RH5 and its role in invasion is a recent evolutionary innovation. To date, all binding studies with RH5 complex components have been performed using the *P. falciparum* orthologues of these proteins. To determine whether the interactions between the components of the RH5 complex were conserved across the different extant *Laverania* species, we expressed the orthologues of P113 and CyRPA from each of the *Laverania* parasites as bait proteins and tested their ability to bind RH5 prey proteins using the AVEXIS assay. Within each species, we observed that both the RH5-P113 and RH5-CyRPA interactions could be robustly detected (Fig 3B). We also asked whether interactions between the orthologues of these proteins from different parasite species could interact by systematically testing them against one another using the AVEXIS assay. We observed that in general, they could, with positive binding observed for all RH5-P113 and RH5-CyRPA combinations, with the exception of *P. billcollinsi* RH5, which did not interact with *P. adleri* CyRPA (Fig 3B). We validated these results by quantifying the binding parameters for a subset using SPR (S5 Fig). These data demonstrate that interactions between the RH5 complex components are conserved within the *Laverania*, and because the RH5-CyRPA interaction is conserved between species, suggest that the co-transfer of the linked *cyrpa* gene was not critical for the zoonotic species jump to humans.

The ability of the introgressed RH5 protein to interact with both gorilla and human basigin was likely to be a critical molecular property in the host switch to human; however, *P. falciparum* is restricted to humans and *P. falciparum* RH5 only binds human basigin [21]. What, then, were the adaptive changes in the RH5 sequence that likely led to human specialization? There are six amino acid differences between the introgressed and the reference (3D7 strain) *P. falciparum* RH5 sequence, some of which are close to the basigin binding site (Fig 4A and S6 Fig). To explore the molecular evolutionary changes that led to the loss of gorilla basigin binding, we individually mutated all six residues that differed in the introgressed RH5 sequence to the equivalent residue in the *P. falciparum* RH5 sequence, and binding to human and gorilla basigin was quantified by SPR. As expected, all six protein variants bound human basigin and did so with similar affinities (Fig 4B and Fig 4C). By contrast, the mutation at position 200 resulted in the complete loss of binding to gorilla basigin (Fig 4B and Fig 4C). These results demonstrate that a single residue was required for the specialization of RH5 binding to human basigin.

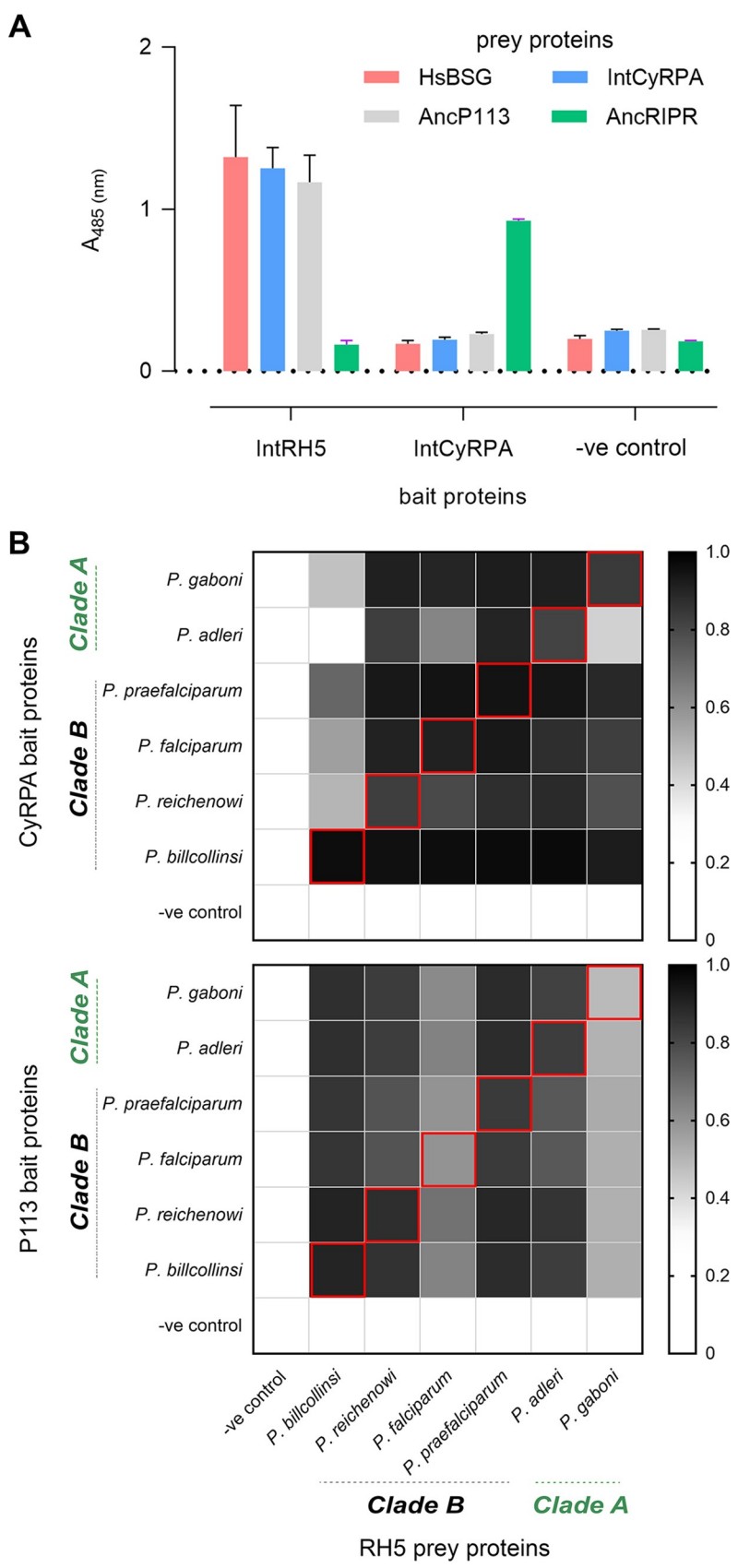

**Fig 3. RH5 complex interactions are conserved across the *Laverania*, including the introgressed RH5. (A)**
Interactions between the introgressed (Int) and ancestral (Anc) *Laverania* RH5 complex components are conserved.
Introgressed RH5 binds introgressed CyRPA and ancestral P113, and introgressed CyRPA binds ancestral RIPR, as
shown by the AVEXIS assay using the named bait and prey proteins. Bars represent means ± SEM; *n* = 3. **(B)** Summary
of the interactions within the *Laverania* RH5 complex. The orthologues of RH5, P113, and CyRPA from the named
*Laverania* species were synthesized, expressed, and systematically tested for binding using the AVEXIS assay. The
RH5-CyRPA (upper panel) and RH5-P113 (lower panel) interactions were conserved within (indicated by red boxes) 
and across all tested species, with the exception of *P. billcollinsi* RH5, which did not interact with *P. adleri* CyRPA. The
scale graphically represents the normalized quantitative readout of the AVEXIS assay but is not expected to provide
relative measures of interaction affinity; values above 0.4 are considered positive. Underlying numerical data can be
found in S1 Data. A single representative of three experiments is shown. Anc, ancestral; AVEXIS, avidity-based
extracellular interaction screening; CyRPA, cysteine-rich protective antigen; HsBSG, human basigin; Int, introgressed;
RH5, reticulocyte-binding protein homologue 5; RIPR, RH5-interacting protein; -ve, negative control.

## Discussion

The discovery that the origins of *P. falciparum* are the *Laverania* has raised questions of what
determines the strict tropism of these parasites for their gorilla and chimpanzee hosts, and,
perhaps more importantly, what were the molecular changes that permitted the zoonotic infec-
tion of humans. Anecdotal observations suggesting a molecular restriction factor in the blood
stages [23], the identification of an introgression event encoding the essential invasion ligand
RH5 [8,9,16], and the specificity of *P. falciparum* RH5 binding to human, but not gorilla or
chimpanzee basigin [21], all implicated a role for this host–parasite receptor–ligand interac-
tion; however, the molecular pathway triggering the zoonotic infection of humans was
unknown.

Our systematic binding studies using the RH5 ligand from the different species within the
*Laverania* demonstrate that not all of these parasites rely on host basigin receptor binding
specificity to govern host tropism. While the RH5 ligand from clade B parasites, *P. billcollinsi*
and *P. reichenowi*, bound only the basigin receptor from their known hosts, RH5 proteins
from the clade A parasites, *P. gaboni* and *P. adleri*, were more promiscuous and could interact
with both the basigin receptor from humans as well as their ape host. This difference provides
a molecular explanation for the ability of the ancestor of *P. falciparum* to breach species barri-
ers by the inter-clade transfer of a promiscuous "clade A" RH5 ligand that can bind both
gorilla and human basigin into an ancestral "clade B" parasite, which likely depended on the
specificity of this receptor interaction for host tropism. This finding therefore challenges the
general view, based on the binding specificity of *P. falciparum* RH5 for human basigin [21],
that the host tropism of all *Laverania* parasites is determined by this same mechanism. Given
the ability of their RH5 protein to bind the basigin receptor from multiple hosts, it is clear that
other factors must determine host tropism for the clade A parasites and *P. praefalciparum*.
Comparative genome analysis of the *Laverania* parasites suggests likely candidates are also
erythrocyte invasion ligands from the Rh and erythrocyte binding antigen families, because
the genes within these families are variably present or absent, are pseudogenes, or have multi-
ple paralogues [8]. It is, however, formally possible that other parasite proteins implicated in
the adhesion to the red blood cell, such as the merozoite surface proteins (MSPs), or proteins
that are important at other stages of the life cycle could be involved.

Coinfections of *Laverania* parasites are frequent in apes [24,25], providing ample opportu-
nity for gene flow between these parasites, and it is highly likely that the founding event encod-
ing *rh5* and *cyrpa* occurred during a coinfection in gorillas. Despite this, comparative genome
analysis revealed clear evidence for few introgression events, suggesting they were either very
infrequent and/or strongly selected against [8]. By showing that the interactions between the
RH5-complex components are conserved across the *Laverania*, we provide an explanation for
why this introgression event was tolerated, because a functional RH5 complex is likely to have

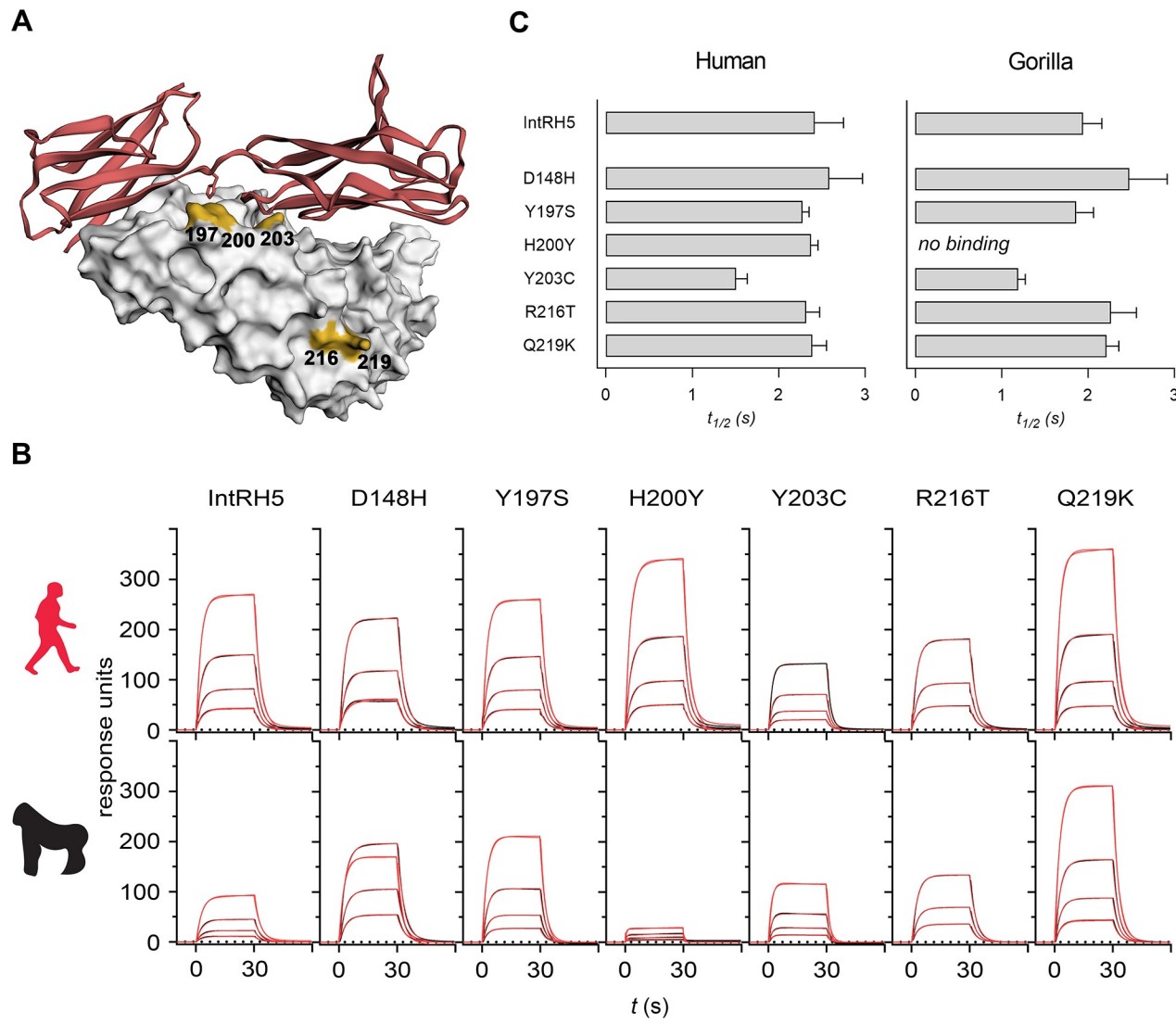

**Fig 4. A single amino acid in RH5 is responsible for human basigin binding specialization. (A)** Structural representation of the RH5–basigin complex, with five of the six residues that differ between the introgressed RH5 and *P. falciparum* RH5 highlighted in yellow. Note that residue 148 was not resolved in the RH5 structure. **(B)** Proteins corresponding to IntRH5 and the six mutations where the amino acids have been mutated to the corresponding residue in the *P. falciparum* RH5 sequence were expressed and purified and the binding to either human or gorilla basigin determined by SPR. Sensorgrams of four 2-fold serial dilutions (600 to 75 nM) are shown, with the raw data shown in black and the fits to a simple 1:1 binding model in red. Representative sensorgrams from two experiments are shown. All mutants bound human basigin with comparable affinity (top row), and one (H200Y) showed no detectable binding to gorilla basigin. **(C)** Calculated interaction half-lives ($t_{1/2}$) of all mutants to human and gorilla basigin. Bars represent means ± SEM from at least three different analyte concentrations. Underlying numerical data can be found in S1 Data. IntRH5, introgressed ancestral RH5; RH5, reticulocyte-binding protein homologue 5; SPR, surface plasmon resonance.

formed immediately following gene transfer. We can consequently infer that any future gene transfer events involving RH5 complex components are also likely to result in viable recombinant parasites. The finding that there is little binding specificity between the RH5 and CyRPA orthologues from different species also demonstrates that there was no need to preserve binding compatibility between the transferred RH5 and CyRPA proteins, which would have limited any shortening of a possibly larger introgressed region by recombination [9]; rather, our analysis suggests that the cotransfer of *cyrpa* was not critically required.

One intriguing aspect of *Laverania* host–parasite relationships is the restriction of each parasite to a specific host, and it appears that host specialization has occurred multiple times following the clade A/B split [8]. The restriction of *P. falciparum* to humans is very likely to have involved changes in the RH5 sequence so that it has become specific for the human basigin receptor [21]. Recent sequence comparisons have identified several codons that may have undergone positive selection in *rh5* during the evolution of the *Laverania*, including three that encode amino acids involved in basigin binding [26,27]. Of these, the residue at position 197 is likely to have changed subsequent to the host switch from gorillas to humans [26], although approximately half of the sequenced Southeast Asian *P. falciparum* samples have retained the ancestral tyrosine at this position (S2 Table). In addition, we did not observe any overt difference in binding affinity to human or gorilla basigin between the ancestral protein and the protein with the mutated residue. While the Y197 allele is more frequent in Southeast Asia, the introgressed H148 allele is present in approximately 18% of *P. falciparum* isolates, and the Y203 allele is dominant globally, being observed in approximately 86% of sequenced isolates. Conversely, the introgressed RH5 residues H200, R216, and Q219 have not been identified in extant *P. falciparum* populations (S2 Table). Here, we have shown that a single substitution at position 200, which is also located within the receptor contact interface, can explain the loss of gorilla basigin binding and thereby restrict *P. falciparum* RH5 to the human receptor. Analysis of *P. falciparum* genomes from different geographical locations reveals that this allele is fixed in all sequenced isolates [28]. Currently, there is no evidence that fixation of this substitution was positively selected due to enhanced receptor binding because the affinity for human basigin was almost identical between the ancestral and derived allele, and there is no signal of positive selection at this site using interspecific sequence comparisons [26,27]. It is therefore possible that this substitution evolved neutrally, perhaps as a result of a parasite population bottleneck approximately 5,000 years ago [8].

While it seems clear that the horizontal transfer of *rh5* was an important event in the origin of *P. falciparum*, there has been discussion as to whether this was a discrete event, whereby a single parasite acquired the ability to exclusively infect humans, or whether it involved a more gradual adaptation in which genetic exchange between the parasite populations in gorillas and humans continued after the introgression event [8,9]. Genome analysis suggests that there was gene flow during this period of adaptation, which eventually ended, causing parasite speciation [8]. Our results provide a molecular mechanism for this multistep process by showing that the introgressed RH5 protein could permit the infection of both gorillas and humans, followed by a mutation at position 200 in RH5 that restricted the parasite to humans. This pathway of adaptation provides an explanation for the presence of ancient dimorphic loci in *P. falciparum* isolates that predate the *P. falciparum*/*P. praefalciparum* last common ancestor [8], and does not invoke the need for a genetic change that would simultaneously permit both switching and restriction to the human host. Finally, malaria remains a major global health problem, and elucidating the molecular events that likely led to the origin of *P. falciparum* and their functional effects is of interest not only to understand how this parasite came to cause such a devastating infection of man but also to assess the likelihood, by informed parasite sequence surveillance, of future zoonotic events from the natural reservoir of related parasites.

## Materials and methods

### Recombinant protein expression and purification

*P. falciparum* RH5 and its orthologues were expressed as soluble recombinant Cd4-tagged proteins that additionally contained a C-terminal 6-his tag for purification, as previously described [16]. The accession numbers used for proteins sequences were RH5 (PF3D7_0424100), P113

(PF3D7_1420700), CyRPA (PF3D7_0423800), and RIPR (PF3D7_0323400). The additional six *Laverania* RH5 orthologues and introgressed RH5 sequences were synthesized by gene synthesis (Geneart, Germany) essentially as described [29]. The *P. blacklocki* genome sequence was derived from just a single isolate that required selective whole genome amplification, resulting in a less complete genome sequence [8]. The expressed recombinant protein corresponding to the *P. blacklocki* RH5 orthologue was prone to the formation of aggregates and did not show biological activity by binding gorilla basigin or *P. blacklocki* CyRPA or P113, leading us to conclude that there may be errors in the sequence; *P. blacklocki* RH5 was therefore excluded from further analysis. To make mutations in the introgressed RH5 sequence, PCR primers were designed with the intended nucleotide change and site-directed mutagenesis performed using KOD Hot Start DNA polymerase (EMD Chemicals, San Diego, CA), as per the manufacturer's instructions. The entire ectodomains of human, chimpanzee, and gorilla basigin were expressed as enzymatically monobiotinylated proteins, as described [16,21]. Genome comparisons of archaic humans, including Neanderthals and Denisovians, showed no polymorphisms affecting the basigin protein sequence in comparison to the *Homo sapiens* reference sequence, suggesting that the host basigin sequence is very likely to have been conserved in hominins over the timescale of the divergence of *P. falciparum*. All proteins were expressed as secreted proteins by transient transfection of the human HEK293E cell line grown in suspension as described [30]. When required, proteins were purified from spent tissue culture media using $Ni^{2+}$-NTA resin using an AKTA pure instrument (GE Healthcare, Chicago, IL) [31].

## SPR

SPR studies were performed using a Biacore 8K instrument (GE Healthcare, Chicago, IL). Biotinylated bait proteins were captured on a streptavidin-coated sensor chip (GE Healthcare, Chicago, IL). Approximately 400 RU of the negative control bait (biotinylated rat Cd4d3+4) were immobilized in the flow cell used as a reference and approximate molar equivalents of the query protein immobilized in other flow cells. Purified analyte proteins were separated by size exclusion chromatography on a Superdex 200 Increase 10/300 column (GE Healthcare, Chicago, IL) in HBS-EP just prior to use in SPR experiments to remove any protein aggregates that might influence kinetic measurements. Increasing concentrations of purified proteins were injected at 100 μL/minute to determine kinetic parameters or at 20 μL/minute for equilibrium measurements. The surface was regenerated with a pulse of 2 M NaCl at the end of each cycle. Duplicate injections of the same concentration in each experiment were superimposable, demonstrating no loss of activity after regenerating the surface. Both kinetic and equilibrium binding data were analyzed in the manufacturer's Biacore 8K evaluation software version 1.1 (GE Healthcare, Chicago, IL). Equilibrium binding measurements were taken once equilibrium had been reached, using reference-subtracted sensorgrams. Both the kinetic and equilibrium binding were replicated using independent protein preparations of both ligand and analyte proteins. All experiments were performed at 37 °C in HBS-EP (10 mM HEPES, 150 mM NaCl, 3 mM EDTA, 0.05% v/v P20 surfactant).

## AVEXIS

Interaction screening was carried out as previously described [30]. Briefly, both bait and prey protein preparations were normalized to activities that have been previously shown to detect transient interactions (monomeric half-lives less than 0.1 second) with a low false positive rate [15]. Biotinylated baits that had been dialyzed against PBS were immobilized in the wells of a streptavidin-coated 96-well microtiter plate (NUNC, Denmark). Normalized preys were added, incubated for 1 hour at room temperature, washed three times in PBS/0.1% Tween-20

and once in PBS, after which 125 μg/mL of nitrocefin (Cayman Chemicals, Ann Arbor, MI) was added and absorbance values measured at 485 nm on a Fluostar Optima (BMG laboratories, Aylesbury, United Kingdom).

### Ancestral sequence reconstruction

The reconstruction of the ancestral sequence of *P. falciparum* and *P. praefalciparum* for each gene (*rh5*, *cyrpa*, *p113*, and *ripr*) was performed using FastML (http://fastml.tau.ac.il). FastML is a program using maximum likelihood and empirical Bayes methods to reconstruct the ancestral characters and compute the posterior probabilities of each ancestral character state. An alignment performed using the translation alignment program available in Geneious v.11. (Biomatters, New Zealand) and including the sequences of all known *Laverania* species (*P. falciparum*, *P. praefalciparum*, *P. reichenowi*, *P. billcollinsi*, *P. blacklocki*, *P. gaboni*, *and P. adleri*) was provided for each gene. A maximum likelihood tree for each gene was estimated de novo by the program using a General Time Reversible (GTR) model of molecular evolution. For each sequence, the ancestral reconstruction was performed using two methods: the marginal and the joint reconstruction methods. For the first method, ancestral characters were inferred at the specific node of interest (the node joining *P. falciparum* and *P. praefalciparum*), while for the second one, all internal node sequences were simultaneously inferred. In all our analyses, both methods provided identical ancestral sequences for the node of interest.

### RH5 cell binding assay

HEL cells were cultured and probed for binding with RH5 orthologues that were multimerized around streptavidin-phycoerythrin (PE) (Biolegend, San Diego, CA) as described [22]. Briefly, each RH5 orthologue was expressed as a secreted biotinylated monomeric protein with the Cd4(d3+4) tag in HEK293 cells [16]. Spent tissue culture supernatant was dialyzed, serially diluted, and incubated with a fixed streptavidin-PE concentration (1 μg/mL) before transferring complexes to a streptavidin-coated plate (NUNC, Denmark) to determine presence of excess RH5 monomer by ELISA, using OX68 (mouse anti-rat Cd4(d3+4)). To make avid tetramers without excess RH5 monomers for the cell binding assay, the minimal dilution of each RH5 protein was used that demonstrated complete capture by the free streptavidin. Each RH5 tetramer-PE (at 1 μg/mL streptavidin-PE) was incubated with $1 \times 10^6$ cells in a total volume of 100 μL for 30 minutes. The cells were washed with PBS and analyzed by flow cytometry. The control anti-CD59 (clone BRIC 229) and blocking anti-basigin Ab-1 [32] antibodies were incubated with the cells for 30 minutes at 10 μg/mL prior to the RH5 tetramer-PE addition.

### Structural representation and modeling

PyMol 2.0 software (Schrodinger, New York, NY) was used for structural representation and modeling (PDB file 4u0q). The illustrative graphic of the RH5 complex was created using PDB files 4u0q, 5ezo, and 6mpv.

### Supporting information

**S1 Fig. Schematic representation of Chromosome 4 in and around the introgressed region.** Each DNA strand is represented by gray bars, and open reading frames encoding the named protein products are colored. The scale indicates the equivalent position in the *P. falciparum* 3D7 reference genome. The phylogenetic topologies calculated for the introgressed and flanking sequences are provided, illustrating the extent and origin of the introgressed region. (TIF)

**S2 Fig. Introgressed RH5 and RH5 proteins from extant *Laverania Plasmodium* spp.** The indicated *Laverania* RH5 proteins were expressed in HEK293 cells as secreted recombinant proteins with a Cd4(d3+4)-His$^{6+}$ tag and purified by immobilized Ni$^{2+}$ ion chromatography. Proteins were resolved by SDS-PAGE under reducing conditions and stained with Coomassie brilliant blue. Expected molecular masses: introgressed RH5, 84.9 kDa; *P. gaboni*, 85.2 kDa; *P. adleri*, 84.9 kDa; *P. praefalciparum*, 85.7 kDa; *P. falciparum*, 84.7 kDa; *P. reichenowi*, 82.5 kDa; *P. billcollinsi*, 84.9 kDa. Original unprocessed gels can be found in S1 Data. Cd4(d3+4)-His$^{6+}$, Ig-like domains 3 and 4 of rat CD4; RH5, reticulocyte-binding protein homologue 5.
(TIF)

**S3 Fig. RH5 from an ancient inter-clade introgression event in *Laverania* parasites binds a HEL cell line in a basigin-dependent manner.** Binding specificities of *Laverania* parasite RH5 proteins with human basigin were confirmed by cell binding experiments. The introgressed RH5 protein was expressed as an enzymatically monobiotinylated protein, purified, and clustered around a streptavidin-PE conjugate to create an avid RH5 labeled binding probe before presenting to basigin-expressing HEL cells. Specificity was demonstrated by showing that RH5 probe binding activity was abolished by preincubating the cells with an anti-basigin mAb that blocks RH5 binding (Ab1—blue histograms) compared with a cell-binding isotype-matched anti-CD59 mAb (red). Control is streptavidin-PE alone (gray). Summary numerical data are provided in S1 Data; gating strategy and original .fcs files in S2 Data. Ab1, anti-basigin mAb; HEL, human erythroid-like; mAb, monoclonal antibody; PE, phycoerythrin; RH5, reticulocyte-binding protein homologue 5.
(TIF)

**S4 Fig. Introgressed RH5 is able to directly interact with introgressed CyRPA and ancestral P113.** SPR traces showing that the introgressed RH5 protein is able to directly interact with the introgressed CyRPA (**A**) and with the known RH5 binding site in the N terminus of P113 (**B**). Both the introgressed CyRPA and N terminus of the ancestral P113 were expressed as soluble enzymatically monobiotinylated proteins and 800 RU and 600 RU were captured, respectively, on the surface of a streptavidin-coated sensor chip. Serial dilutions of purified introgressed RH5 were injected at 100 μL/minute over IntCyRPA (full-length introgressed RH5) and P113Nt (N terminus of introgressed RH5), respectively, and the biophysical binding parameters of the interaction calculated by fitting the binding data (black) to a simple 1:1 binding model (red). Underlying numerical data can be found in S1 Data. CyRPA, cysteine-rich protective antigen; IntCyRPA, introgressed ancestral CyRPA; P113Nt, P113 N-terminal domain; RH5, reticulocyte-binding protein homologue 5; SPR, surface plasmon resonance.
(TIF)

**S5 Fig. The interactions between the RH5 complex components are conserved across the *Laverania* subgenus. (A)** Representative SPR sensorgrams quantifying the RH5-CyRPA (left panel) and RH5-P113 (right panel) interactions used to calculate the summary data shown in (**B**). In this example, serial dilutions of *P. falciparum* RH5 were used as the analyte with enzymatically monobiotinylated *P. falciparum* CyRPA and P113 immobilized on a streptavidin-coated sensor chip. Biophysical binding parameters were calculated by fitting the raw binding data (black) to a simple 1:1 binding model (red). (**B**) A summary of affinity measurements between *P. falciparum* (*Pf*) and *P. praefalciparum* (*Pp*) RH5 and *P. falciparum*, *P. praefalciparum*, and *P. adleri* CyRPA and P113. The equilibrium dissociation constants ($K_D$) for each interaction was calculated from 1:1 fits to the SPR binding data and plotted. Bars represent means ± s.e.m. from at least five different analyte concentrations. Underlying numerical data can be found in S1 Data. CyRPA, cysteine-rich protective antigen; *Pf*, *P. falciparum*; *Pp*, *P.*

*praefalciparum*; RH5, reticulocyte-binding protein homologue 5; SPR, surface plasmon resonance.
(TIF)

**S6 Fig. Alignment of the RH5 sequence obtained from the extant species and the estimated ancestral sequence.** Black bars indicate a difference with the ancestral sequence. The red dots indicate a nonsynonymous substitution in *P. falciparum* and *P. praefalciparum*. RH5, reticulocyte-binding protein homologue 5.
(TIF)

**S1 Table. Table showing the confidence at each residue of the RH5 complex components, RH5, CyRPA, RIPR, and P113.** CyRPA, cysteine-rich protective antigen; RH5, reticulocyte-binding protein homologue 5; RIPR, RH5-interacting protein.
(XLSX)

**S2 Table. Publicly available MalariaGEN data showing non-reference allele frequencies (NRAF) in the *P. falciparum* population for RH5.** The introgressed H148 allele is present in 18% of *P. falciparum* isolates, while the Y197 allele dominates in Southeast Asia. The Y203 allele is dominant globally (86% of sequenced isolates), making the 3D7 strain unrepresentative for this position. The H200, R216, and Q219 present in the calculated introgressed RH5 sequence have not been detected in extant sequenced *P. falciparum* populations. CAF, Central Africa; EAF, East Africa; ESEA, East Southeast Asia; FST, population differentiation statistic; MAF, global allele frequency; NRAF, non-reference allele frequencies; PNG, Papua New Guinea; SAM, South America; SAS, South Asia; WAF, West Africa; WSEA, West South East Asia; RH5, reticulocyte-binding protein homologue 5.
(TIF)

**S1 Data. Contains data pertaining to** Fig 1C, Fig 2A, Fig 2B, Fig 3A, Fig 3B, Fig 4B, Fig 4C, S2 Fig, S3 Fig, S4 Fig, S5 Fig.
(XLSX)

**S2 Data. Contains flow cytometry gating strategy and original .fcs files for flow cytometry data shown in** Fig 2B **and** S3 Fig.
(ZIP)

## Acknowledgments

We thank Thomas Otto for sequences.

## Author Contributions

**Conceptualization:** Francis Galaway, Franck Prugnolle, Gavin J. Wright.

**Data curation:** Francis Galaway.

**Formal analysis:** Francis Galaway, Ryan Yu, Anastasia Constantinou, Gavin J. Wright.

**Funding acquisition:** Franck Prugnolle, Gavin J. Wright.

**Investigation:** Francis Galaway, Ryan Yu, Anastasia Constantinou, Franck Prugnolle, Gavin J. Wright.

**Methodology:** Francis Galaway, Franck Prugnolle, Gavin J. Wright.

**Project administration:** Gavin J. Wright.

**Supervision:** Francis Galaway, Franck Prugnolle, Gavin J. Wright.

**Writing – original draft:** Francis Galaway, Gavin J. Wright.

**Writing – review & editing:** Franck Prugnolle.

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
