## [Editor Report · Decision Letter 0]

28 Jun 2019

Dear Dr Prugnolle, 

Thank you for submitting your manuscript entitled "Resurrection of the ancestral RH5 invasion ligand provides a molecular explanation for the origin of P. falciparum malaria in humans" for consideration as a Research Article by PLOS Biology following our presubmission discussion.

Your manuscript has now been evaluated by the PLOS Biology editorial staff as well as by an academic editor with relevant expertise and I am writing to let you know that we would like to send your submission out for external peer review.

**Important**: Please also see below for further information regarding completing the MDAR reporting checklist. The checklist can be accessed here: https://plos.io/MDARChecklist

Please re-submit your manuscript and the checklist, within two working days, i.e. by Jun 30 2019 11:59PM.

Kind regards,

Lauren A Richardson, Ph.D

Senior Editor

PLOS Biology

INFORMATION REGARDING THE REPORTING CHECKLIST:

PLOS Biology is pleased to support the "minimum reporting standards in the life sciences" initiative (https://osf.io/preprints/metaarxiv/9sm4x/). This effort brings together a number of leading journals and reproducibility experts to develop minimum expectations for reporting information about Materials (including data and code), Design, Analysis and Reporting (MDAR) in published papers. We believe broad alignment on these standards will be to the benefit of authors, reviewers, journals and the wider research community and will help drive better practise in publishing reproducible research. 

We are therefore participating in a community pilot involving a small number of life science journals to test the MDAR checklist. The checklist is intended to help authors, reviewers and editors adopt and implement the minimum reporting framework. 

IMPORTANT: We have chosen your manuscript to participate in this trial. The relevant documents can be located here:

MDAR reporting checklist (to be filled in by you): https://plos.io/MDARChecklist

**We strongly encourage you to complete the MDAR reporting checklist and return it to us with your full submission, as described above. We would also be very grateful if you could complete this author survey:

https://forms.gle/seEgCrDtM6GLKFGQA

Additional background information:

Interpreting the MDAR Framework: https://plos.io/MDARFramework

Please note that your completed checklist and survey will be shared with the minimum reporting standards working group. However, the working group will not be provided with access to the manuscript or any other confidential information including author identities, manuscript titles or abstracts. Feedback from this process will be used to consider next steps, which might include revisions to the content of the checklist. Data and materials from this initial trial will be publicly shared in September 2019. Data will only be provided in aggregate form and will not be parsed by individual article or by journal, so as to respect the confidentiality of responses. 

Please treat the checklist and elaboration as confidential as public release is planned for September 2019.

We would be grateful for any feedback you may have.

---

## [Decision Letter · Decision Letter 1]

31 Jul 2019

Dear Dr Prugnolle,

Thank you very much for submitting your manuscript "Resurrection of the ancestral RH5 invasion ligand provides a molecular explanation for the origin of P. falciparum malaria in humans" for consideration as a Research Article at PLOS Biology. Your manuscript has been evaluated by the PLOS Biology editors, an Academic Editor with relevant expertise, and by several independent reviewers.

In light of the reviews (below), we are pleased to offer you the opportunity to address the comments from the reviewers in a revised version that we anticipate should not take you very long. Of particular note, the reviewers request a more robust introduction and a more nuanced discussion of the results. Rev #2 also raises an important point about variation within Plasmodium sequences. 

Your revisions should address the specific points made by each reviewer. Please submit a file detailing your responses to the editorial requests and a point-by-point response to all of the reviewers' comments that indicates the changes you have made to the manuscript. In addition to a clean copy of the manuscript, please upload a 'track-changes' version of your manuscript that specifies the edits made. This should be uploaded as a "Related" file type. You should also cite any additional relevant literature that has been published since the original submission and mention any additional citations in your response. 

Before you revise your manuscript, please review the following PLOS policy and formatting requirements checklist PDF: http://journals.plos.org/plosbiology/s/file?id=9411/plos-biology-formatting-checklist.pdf. It is helpful if you format your revision according to our requirements - should your paper subsequently be accepted, this will save time at the acceptance stage.

Please note that as a condition of publication PLOS' data policy (http://journals.plos.org/plosbiology/s/data-availability) requires that you make available all data used to draw the conclusions arrived at in your manuscript. If you have not already done so, you must include any data used in your manuscript either in appropriate repositories, within the body of the manuscript, or as supporting information (N.B. this includes any numerical values that were used to generate graphs, histograms etc.). For an example see here: http://www.plosbiology.org/article/info%3Adoi%2F10.1371%2Fjournal.pbio.1001908#s5.

For manuscripts submitted on or after 1st July 2019, we require the original, uncropped and minimally adjusted images supporting all blot and gel results reported in an article's figures or Supporting Information files. We will require these files before a manuscript can be accepted so please prepare them now, if you have not already uploaded them. Please carefully read our guidelines for how to prepare and upload this data: https://journals.plos.org/plosbiology/s/figures#loc-blot-and-gel-reporting-requirements.

Upon resubmission, the editors assess your revision and assuming the editors and Academic Editor feel that the revised manuscript remains appropriate for the journal, we may send the manuscript for re-review. We aim to consult the same Academic Editor and reviewers for revised manuscripts but may consult others if needed.

We expect to receive your revised manuscript within one month. Please email us (plosbiology@plos.org) to discuss this if you have any questions or concerns, or would like to request an extension. At this stage, your manuscript remains formally under active consideration at our journal; please notify us by email if you do not wish to submit a revision and instead wish to pursue publication elsewhere, so that we may end consideration of the manuscript at PLOS Biology.

When you are ready to submit a revised version of your manuscript, please go to https://www.editorialmanager.com/pbiology/ and log in as an Author. Click the link labelled 'Submissions Needing Revision' where you will find your submission record. 

Sincerely,

Lauren A Richardson, Ph.D

Senior Editor

PLOS Biology

Reviews

Reviewer #1: 

Galaway and co-authors describe their efforts to recreate and phenotype an ancestral version of a protein (Rh5) that mediates red blood cell invasion by malaria parasites and is thought to have been a crucial player in the origin of P. falciparum as a human parasite through an introgression event from a gorilla parasite lineage. This approach taken by the authors to express extant and ancestral versions of the Rh5 protein is elegant, and the authors employ both SPR and cell-based assays to produce robust phenotype profiles. This work is an important addition to the exciting recent advances in understanding the origins of human malaria and the role of red blood cell invasion in host tropism.

The manuscript is clearly written and the authors’ conclusions are generally well supported. I have the following questions and suggestions for minor clarifications:

1) The ancestral gene reconstruction for IntRH5 is included in a supplemental table, but a cursory description of how divergent it is from extant Pfal/Padleri/Ppraefalciparum sequences in the main text would be useful. How many amino acid subs from extant species? It could be useful, for example, to see a small tree showing the subs that have subsequently occurred on the Pfal/Pprae lineages since introgression to understand the rough magnitude of divergence that has subsequently occurred.

2) On a similar note, near the end of Results the authors mention 6 AA differences between the introgressed and reference 3D7 assembly sequence for Rh5. It would be helpful to clarify here whether 3D7 is expected to reflect ancestral or derived alleles at these positions for P. falciparum, given that Rh5 is polymorphic in contemporary parasite populations (albeit lowly relative to other invasion genes).

3) The Discussion does not remark upon an observation one can make by comparing Figures 2A and 4B: that contemporary Pprae Rh5 is better at binding both human and gorilla basigen than the inferred introgressed allele (IntRH5). IntRH5 appears to have been a fairly poor binder to gorilla basigen, in fact, suggesting that subsequent adaption within the Ppra lineage may have been necessary to achieve more effective infection of gorillas following an introgression event and selective sweep driven by the capacity to successfully infect humans. This is a surprising finding, perhaps indicating that the introgression event subsumed a presumed pre-existing chimp host tropism in the Pf/Pprae ancestral lineage due to the evolutionary advantage arising from the capacity to infect humans.

---------------

Reviewer #2: 

In their manuscript entitled “Resurrection of the ancestral RH5 invasion ligand provides a molecular explanation for the origin of P. falciparum malaria in humans” the authors seek to show how Plasmodium parasites may have switched hosts, evolving from a gorilla-specific parasite to one that can infection humans (Plasmodium falciparum). To do this, they “resurrect” the introgressed ancestral RH5 gene, which encodes the ligand which enables erythrocyte invasion by binding the basigin receptor, and conduct protein interaction assays to test binding. By doing so, they show that the ancestral P. falciparum RH5 gene can bind various receptors. Furthermore, they show that specificity to one species, or more precisely loss of specificity to a species, can arise from point mutations in the sequence. The study provides interesting molecular data to support the hypothesis that human specificity evolved via changes in rh5.

The manuscript is well written and the authors provide compelling biochemical binding data. On the other hand, tests with recombinant proteins do not really tell you whether the mutations will have the same function in the context of an infection with other proteins present. Should one try swapping expression of P. falciparum rh5 with ancestral rh5 using a conditional knockdown system (for example, with a tet aptamer system, degradation domain), and test to see if this changes affinity for human erythrocytes? Such experiments are difficult and may be outside the scope of this study, however. 

In addition, the authors focus on a single 3D7 sequence and there may be other rh5 and basigin gene variants present in their respective human and parasite populations. The authors state that there are only six nonsynonymous changes between the P. falciparum rh5 and rh5 introgressed sequence and yet there seem to be more than six nonsynonymous rh5 mutations within existing P. falciparum populations (e.g. those present in PlasmoDB), and some of these variants are present in the introgressed rh5 variant (positions 148, 197, 203). Would the authors’ results be different if the authors used different P. falciparum sequences? Although the 3D7 strain is the most heavily studied, it may or may not be a good representative for studies on species evolution. Do any of the phylogenetic trees change if one uses different strains?

Minor points: 

The introduction is somewhat brief and to understand this, additional background reading is needed. If length is not a constraint, this could be expanded. For example, are there additional known functions of P113, RIPR, CyRPA (only presented interactions with RH5, but what does interaction entail); knock-out studies, overexpression studies, conserved regions/homology between species?

Please elaborate on how rh5 might be druggable, since this is mentioned. Would this be by disrupting protein interactions? Are there many drugs that work this way? Would an rh5 drug work against many different rh5 isolates?

Briefly state how was the sequence of the ancestral RH5 determined in the text to aid readability.

Briefly elaborate on the surface binding assay in the results. The reader currently needs to go to the methods and other publications to understand this. 

Briefly explain what an introgressed sequence is. 

The authors write, “Interestingly, the introgressed fragment encompassed two genes encoding RH5 complex components: RH5 and CyRPA, which are genetically closely linked (Fig. 1A).” but there seems to be no evidence provided in Fig 1A that RH5 and CyRPA are genetically closely linked. How many bases separate the two? Some sort of diagram or specifics would help the reader understand what is happening. 

Discussio: In the discussion of mutating residues, would be good consider that there are different variants for some of these residues in existing population genomic datasets for P. falciparum. 

Discussion: Mention if host tropism could be determined by proteins other than the RH and EBA family proteins?

Figure 1A: what do blue, black, red figures refer to (can be inferred, but better explicit)

Figure 1A: A better description of how the dendrogram was created would be useful. Could a reference be provided?

Figure 1B: Rh5 is drawn so the N- and C-terminal parts are separate, which is confusing. As drawn it also looks like there is no contact between the erythrocyte and the parasite, forcing one to go look at the existing literature. Perhaps a schematic diagram could be given?

Figure 3B: Include clade information

Figure 4A: where is the binding region between RH5 and basigin? Can the authors circle or highlight this region?

Please provide Kd values plus error bars for Figure 2B. 

It would be useful if the authors included the Plasmodium systematic names for proteins that are discussed in Figures and text, given that there can sometimes be multiple names in the literature.

Figure S1: title is redundant with text. 

Figure 2 and Table S1. It would be useful to comment that the results are from flow cytometry. The X axis could read PE fluorescence of flow sorted HEL cells as well to make the data clearer. A diagram of the experimental setup would be even better.

---------------

Reviewer #3: 

This very interesting article. Particularly the fact that P. gaboni shows promiscuous affinity whereas P. blacklocki that also is found in gorillas did not. Could be possible that the so-called clade B is indeed a group of parasites that can infect chimpanzees? Please clarify. 

Nevertheless, the invasion of P. falciparum is a process that we are just starting to understand and there could be other mechanisms involved. Particularly the role of the Rh5–CyRPA–Ripr complex. I suggest checking Wang et al. 2019 Nature 565: 118–121 and discuss their results in that context. I also suggest checking Volz et al 2016 Cell Host Microbe, 20, 60–71. It will be also interesting to discuss, even as a speculation (the authors may not have a way of testing), the role that antibodies may have in inhibiting the process (Healer et al 2019 Cellular Microbiology 21: e13030) and suggest that such experiments could be of interest. Sometimes less efficient binding still suffices to mediate invasion. Although the information is valuable, it may not tell the complete story so the manuscript will improve by adding a more critical discussion of the findings.

---

## [Editor Report · Decision Letter 2]

3 Sep 2019

Dear Dr Wright,

Thank you for submitting your revised Research Article entitled "Resurrection of the ancestral RH5 invasion ligand provides a molecular explanation for the origin of P. falciparum malaria in humans" for publication in PLOS Biology. 

The Academic Editor and I have now assessed your revised manuscript and we're delighted to let you know that we're now editorially satisfied with your manuscript. We will publish your study, assuming you are willing to make the final edits to meet our production requirements. Congratulations!

Before we can formally accept your paper and consider it "in press", we also need to ensure that your article conforms to our guidelines. A member of our team will be in touch shortly with a set of requests. As we can't proceed until these requirements are met, your swift response will help prevent delays to publication.

Please note that you may have the opportunity to make the peer review history publicly available. The record will include editor decision letters (with reviews) and your responses to reviewer comments. If eligible, we will contact you to opt in or out.

Sincerely,

Lauren A Richardson, Ph.D

Senior Editor

PLOS Biology

DATA POLICY:

Regardless of the method selected, please ensure that you provide the individual numerical values that underlie the summary data displayed in the following figure panels: (e.g. Figs. 3AB, 4C, S5B), as they are essential for readers to assess your analysis and to reproduce it. Please also ensure that figure legends in your manuscript include information on where the underlying data can be found.

For figures containing FACS data, we ask that you provide FCS files and a picture showing the successive plots and gates that were applied to the FCS files to generate the figure.

For manuscripts submitted on or after 1st July 2019, we require the original, uncropped and minimally adjusted images supporting all blot and gel results reported in an article's figures or Supporting Information files. We will require these files before a manuscript can be accepted so please prepare them now, if you have not already uploaded them. Please carefully read our guidelines for how to prepare and upload this data: https://journals.plos.org/plosbiology/s/figures#loc-blot-and-gel-reporting-requirements.

---

## [Editor Report · Decision Letter 3]

12 Sep 2019

Dear Dr Wright,

On behalf of my colleagues and the Academic Editor, Andrew Fraser Read, I am pleased to inform you that we will be delighted to publish your Research Article in PLOS Biology. 

Early Version

PRESS 

Kind regards,

Alice Musson

Publication Assistant, 

PLOS Biology

on behalf of

Lauren Richardson,

Senior Editor

PLOS Biology